# Cross-Training Pharmacy Professionals as Community Health Workers

**DOI:** 10.3390/pharmacy11050135

**Published:** 2023-08-27

**Authors:** Paige Somerville, Ryan Lindsay, Elaine Nguyen

**Affiliations:** 1Department of Pharmacy Practice, L.S. Skaggs College of Pharmacy, Idaho State University, Meridian, ID 83642, USA; paigesomerville@isu.edu; 2Department of Community and Public Health, College of Health, Idaho State University, Meridian, ID 83642, USA; ryanlindsay@isu.edu

**Keywords:** community health workers, community health services, pharmacies, training, certification

## Abstract

Community health workers (CHWs) are health professionals that connect the populations they serve to health services. They play a valuable role in assessing patients’ needs, linking patients with health and human resources, educating patients toward achieving optimal health, and advocating for their patients to have accessible resources to improve their health and wellbeing. Pharmacies are readily accessible and frequently utilized health locations that could employ CHWs. We describe a program to cross-train pharmacy professionals as CHWs. Pharmacy professionals were recruited to sign up for a 13-week CHW core competencies course that was offered in partnership with the state university Continuing Education Workforce Training. From March 2022 to June 2023, 23 pharmacy professionals completed the course. Post-course and program evaluations were completed by 10 participants, and they showed positive responses to their participation in the program. Participants appreciated learning the role of a CHW, and noted increased skills, and awareness of additional tools and resources. Participants reported 47 referrals to chronic disease programs and community resources. The program pilot results show successful partnerships for cross-training pharmacy professionals as CHWs. Cross-trained pharmacy professionals provide benefits to the communities they serve, by being a valuable resource for their patients.

## 1. Introduction

Community health workers (CHWs) are health professionals that connect the populations they serve to health services. Many patients face barriers to navigating the healthcare system, and CHWs help by providing resources that patients may not know are available. CHWs play a valuable role in their communities, by assessing patients’ needs, linking patients with health services and resources, educating patients toward achieving optimal health, and advocating for patient resources, to improve health and wellbeing [1,2]. Some of the referral resources that CHWs can provide to their patients include chronic disease management programs, such as pre-diabetes, diabetes, and lifestyle change/weight management programs. CHWs are seen as a bridge that connect their communities with health and social resources.

In Idaho, there are 410 CHWs, according to the latest data (May 2022) from the U.S. Bureau of Labor Statistics [3]. The optimal number of CHWs in a community is not currently known, and yet 38 out of 44 counties in Idaho are designated as health profession shortage areas [4], and estimates of CHWs in community-based organizations are difficult to obtain. Recent HRSA funding [5] has been dedicated to bolstering the CHW workforce, and the number of available training programs, as a priority, as CHWs are increasingly viewed as a crucial, underutilized, and underappreciated component of high-functioning health systems [6].

CHWs work in numerous community locations, including clinics, hospitals, nursing homes, and pharmacies. Pharmacies are highly accessible places of healthcare, with estimates that ~90% of people live within 5 miles of a community pharmacy [7]. In addition, pharmacies have extended hours compared to other locations, which makes it easier for patients to connect with a CHW at a convenient time that works with their schedule. Previous studies have shown that CHWs can effectively be implemented in the pharmacy setting [2], and that their presence can increase the health education related to medication adherence and disease management that already occurs in the pharmacy setting.

In Idaho, pharmacies may even offer expanded health services, as pharmacists have independent prescriptive authority (i.e., they can prescribe drugs or devices) [8]. Pharmacists are also recognized by Idaho Medicaid as healthcare providers, allowing for a potential reimbursement mechanism in health services [9]. Idaho’s broad pharmacy regulations may offer a unique opportunity for CHW service integration, especially in areas where access to health providers or other health services is limited, such as rural areas.

Pharmacy professionals, including pharmacists, student pharmacists, and pharmacy technicians are a vital asset in the communities they serve. The provision of CHW training to these professionals allows them to serve their communities with additional services that will benefit their patients beyond the typical scope of pharmacy practice. Given the potential synergistic impact of CHWs in the pharmacy setting, a program was created to cross-train pharmacy professionals as CHWs.

## 2. Materials and Methods

This initiative represents a partnership between the Idaho Department of Health and Welfare (IDHW), the state university Community Education Workforce Training (CEWT), and the university departments of Community and Public Health and Pharmacy Practice. In January 2022, IDHW and the state university partnered to implement a pilot project to cross-train pharmacy professionals as CHWs. This pilot was conducted in Idaho, due to the state priorities of the funding partner, IDHW. The goal of this project was to increase the number of trained CHWs in the state. A secondary goal was to expand referrals from CHWs based in pharmacies to CDC-recognized diabetes prevention and management programs and lifestyle change programs for cardiovascular disease (e.g., the Diabetes Prevention Program, Diabetes Self-Management Education and Support, and weight loss); this was to encourage CHWs to immediately utilize their new skills to educate patients and link them with new resources.

### 2.1. CEWT CHW Course

The state university CEWT has provided training on the CHW core competencies since 2016, with those completing the course receiving a certificate of completion. The course was originally developed with IDHW support and CHW working group input, and revised over time. It is facilitated through the state university’s Department of Community and Public Health. The 13-week course (48 contact hours) is currently offered as a hybrid online web-based classroom, and consists of asynchronous (self-paced) modules, as well as three live online classroom sessions, where CHWs practice skills such as presenting a case, discussion, and roleplay skills. The courses are team-taught, with at least one instructor being a current CHW. Each week, students are required to complete online learning, unit assignments, and a quiz. Throughout the course, 19 units or modules are covered (14 required, and 5 health-specific electives) that include the key concepts of public health, outreach, advocacy, community and individual assessments, social determinants of health, health education, navigating insurance, stages of behavior change, and service coordination. The curriculum has been mapped to the C3 project’s CHW competencies and skills, and covers content in each area. In addition to the online coursework, there is a final assignment, for which students interview a current CHW in Idaho, and present their findings to the class about the CHW and their organization, and the impact that they have on their community. The goal of the presentations is to learn about programs that are being implemented in Idaho, broaden students’ network, and help them to understand CHW programs that they can partner with in the future. At the beginning and end of the course, students self-report their self-efficacy for the CHW competencies. The course is offered multiple times throughout the year.

### 2.2. Program Recruitment

Through IDHW, funds were provided for 25 pharmacy professionals, to cover the registration fees for the CEWT CHW course, as well as their time in completing the course ($20–22/h × 48 h = $960–$1056). The exact amount varied by year, with a maximum of 25 potential participants, due to funding. Tuition waivers were also utilized to offset registration costs. Individuals interested in participating in the program had to complete an online form, providing their contact information (including the individual’s name and the pharmacy name), a few sentences on how they might utilize their CHW skills, and an agreement to comply with the program requirements. The program requirements were to: provide at least two referrals to CDC-recognized programs for diabetes or cardiovascular disease, report referral information to the project personnel, and complete the CHW course. The completion of program evaluations was also requested, although not required for funding.

The project personnel created and distributed a recruitment flier describing the program and requirements. The flier was distributed through the Community Pharmacy Enhanced Services Network of Idaho (CPESN-ID) and other project personnel contacts. Project personnel also called select independent community pharmacies in Idaho, to inform them of this opportunity. Lastly, student pharmacists at the state university were invited to apply for program participation via email.

### 2.3. Program Support

Program participants were invited to meet with pharmacy project personnel to discuss the sustainable implementation of their new skills at their worksite; this was optional. To assist with referrals, a Google website was created that contained a list (including the program name, location, contact, and website) of CDC-recognized pre-diabetes, diabetes, and lifestyle change programs. Patient handouts were also available, in color and in black and white. Referral resources were provided by the region. Lastly, the Google website also contained a form for program participants to submit referral information (including the date the referral was made, the program referred to, and details about the referral).

### 2.4. Program Evaluation

Program participants were asked to complete course evaluations as part of the CEWT CHW course. In addition, program-specific questions were also asked that inquired about participants’ preparedness to utilize CHW skills in the pharmacy setting, their likelihood of making patient referrals to CDC-recognized disease management programs, and other general feedback (Table 1).

### 2.5. Other Feedback

Project personnel met with four participants after training, to hear feedback, and learn of how they might use their new skills; these were optional 1:1 meetings with participants. In these meetings, and in open-ended evaluation questions, participants expressed appreciation for the training opportunity. They learned the role of a CHW, and noted increased soft skills, such as motivational interviewing, and awareness of additional tools and resources. One pharmacist participant had already expanded their clinical patient intake forms to capture further details on patients’ social determinants of health; this pharmacist is also pursuing advanced CHW training. One consistent area of improvement noted with the training was organization, and the updating of the online course learning management system.

## 3. Results

Throughout the pilot program, 25 pharmacy professionals were selected to participate in the program. However, two individuals withdrew from the course, resulting in 23 individuals completing the course from March 2022 to June 2023. Four participants were pharmacists, seven were pharmacy technicians, and twelve were student pharmacists.

The post-course evaluation was completed by 10 individuals (Table 1). The responses were generally positive, with the majority of respondents stating that program participation prepared them to: utilize their CHW skills in the pharmacy setting, communicate effectively, be able to identify programs for patients, and advocate for CHW skills and roles in pharmacy settings. Eighty percent of respondents were also somewhat/extremely likely to actively integrate their CHW skills into the pharmacy setting, and build referral relationships with chronic disease programs.

Across all the program participants, 47 referrals were made (Table 2). More than half of the referrals were made to diabetes programs. The majority of referrals (*n* = 41) were passive, i.e., program information provided to the patient or patient representative.

## 4. Discussion

We were able to partner with existing programs/resources to recruit pharmacy professionals and train them as CHWs. While the formal feedback on program participation was limited, the results were generally positive, and also indicated opportunities to continue bolstering the program in subsequent years.

Although CHW roles and competencies have been proposed on the national level, there is no national formalized structured curriculum for training CHWs in the United States [10], nor is there an entity responsible for certifying CHW training programs at a national level. As a consequence, the requirements and cost to become a CHW vary by state, which could impact the degree to which community pharmacies may be able to invest in training and employing CHWs. This variation is substantial from state to state, with some states, such as California, not having any formal training or certification for CHWs, and others, such as Florida, Oregon, and Texas, requiring rigorous training to qualify to become a certified CHW [11,12,13]. Florida has a lengthy curriculum for becoming a certified CHW, the requirements of which include a high school diploma or GED, 500 h of formal experience in communication and education, advocacy, or connecting patients with health resources, 30 h of structured CHW training with an additional 10 h of electives, obtaining three letters of recommendation, passing a written CHW test, participating in 10 h of continuing education each year, and renewing one’s CHW certification every two years [11]. In Oregon, the requirements to become a CHW include the completion of an approved CHW training program, or the provision of documentation proving that the participant acquired 3000 h of traditional CHW experience between 1/1/04 and 6/30/19 [12]. If a participant in Oregon has completed some of the training requirements previously, this previously completed training may be considered toward certification [12]. Texas has structured requirements to become a CHW, including being a resident of Texas, and having completed a 160 h Department of State Health Services (DSHS)-certified CHW training program or 1000 cumulative hours of CHW experience in the past three years [13]. To ensure that the CHW’s experience is applicable for certification, verification will take place with the participant’s supervisor [13]. In addition to the requirements above, Texas also requires the CHW to demonstrate knowledge of eight core competencies in order to become a certified CHW [13]. Unlike other states, Idaho has no official certification or state-endorsed training program.

To some extent, the CHW requirements in different states have been driven by fee-for-service payment models and reimbursement from payers (e.g., Medicare and Medicaid). In Idaho, this may explain the lack of official certification, as there is a desire to limit state spending and Medicaid-reimbursable services. However, some states have still chosen not to pursue formal certification, despite reimbursement incentives, to limit barriers to potential CHWs entering the workforce.

Similarly, Idaho has no certification or continuing education requirements for pharmacy technicians [8]. Pharmacy technicians must register with the Idaho State Board of Pharmacy. The registration requirements include: an age of at least 16 years, government-issued photo identification, proof of high school graduation (or equivalent or higher), and fingerprints [8]. The adoption of more rigorous pharmacy technician or CHW requirements may limit potential personnel in both these professions, in both individual and cross-trained roles. This could be detrimental, given the workforce retention difficulties for both pharmacy technicians and CHWs.

Another challenge in cross-training other healthcare personnel as CHWs is that, while cross-training may allow for some increased competencies of skills in the healthcare staff being cross-trained, these individuals may still feel that their agency would benefit from a full-time CHW [14]. Administrators may feel that cross-training is a sufficient replacement for hiring a full CHW. Balancing this concern with the realities of small budgets and limited available personnel in community-based and rural settings may continue to make cross-training healthcare personnel, such as pharmacy personnel, as CHWs desirable. However, this could also be seen as over-medicalizing or co-opting the CHW profession. A related issue that many CHWs might challenge is the notion that any healthcare provider that works with a community can assume the mantle of a CHW with the acquisition of competency and skills. Having the trust of the community, and the qualities necessary to be successful as a CHW are assets that are hard-earned and difficult to teach. Certainly, some of these qualities overlap with pharmacy personnel, and a careful selection of those that should be cross-trained as CHWs, rather than a one-size-fits-all approach, is warranted.

### 4.1. Changes to CEWT CHW Course

The state university’s CHW training program was initially funded through a Centers for Medicaid and Medicare Services State Innovation grant to the State of Idaho. While it has continued to be offered since this initial funding, it recently received funding from IDHW to update the curriculum, and develop advanced CHW training beyond the core competencies that will allow for more topics and specialization tracks (community- and clinic-based), and experiential learning through rotations. A 2000 h registered CHW apprenticeship program was established between the state university and the Idaho Department of Labor in June 2022 that would allow pharmacies to act as apprenticeship employer sites and that has employer incentives to host apprenticeships. In the next few years, the updated CHW core competencies, advanced CHW training, and CHW apprenticeship program will strengthen CHWs and CHW programs in Idaho, through tiered levels of training and experiential learning. Such tiered training will continue to be available in the form of cross-training options for pharmacy professionals. In addition to these changes, two elective modules specifically for CHWs in the pharmacy setting were offered in fall 2023. These electives are offered through a partnership with CEImpact, and also provide pharmacy-accredited continuing education.

### 4.2. Limitations

There are some limitations to this work. The number of participants in the CHW training program was small, making it difficult to effectively understand the impact of the course. Furthermore, our response rate to the program evaluation questions was limited, with only ten respondents. In addition, the generalizability of this project to other sites and states may be limited, due to funding and state-specific CHW requirements.

While the total time commitment involved in the course was estimated to be 48 h, this can still represent a substantial time demand on pharmacy personnel who may already have time constraints. The course itself was also relatively short (13 weeks), and there are limited long-term data on what happened after the participants received their CHW certification, including how the course content was utilized and implemented in the participants’ careers. Future work may consider looking at the implementation of CHW skills, and the facilitators and barriers to doing so (e.g., did CHW-trained participants have the time and employer support to talk with patients to identify needs and provide follow-up resources?).

Despite these limitations, the integration of cross-trained pharmacy professionals as CHWs appears to be advantageous, given the presence and opportunities of pharmacies in communities.

## 5. Conclusions

Our pilot results show successful partnerships for cross-training pharmacy professionals as CHWs. Cross-trained pharmacy professionals provide benefits to the communities they serve, by being a valuable resource for their patients.

## Figures and Tables

**Table 1 pharmacy-11-00135-t001:** Course evaluation responses (*n* = 10).

Question	Response, *n* (%)				
To What Extent Has Program Participation Prepared You to:					
	Not at all	A little	A moderate amount	A lot	A great deal
Utilize your CHW skills in the pharmacy setting	0 (0)	1 (10)	1 (10)	3 (30)	5 (50)
Communicate effectively to others your skills as a cross-trained pharmacy professional and CHW	0 (0)	1 (10)	1 (10)	3 (30)	5 (50)
Communicate effectively between coworkers, patients, and other healthcare professionals	0 (0)	2 (20)	1 (10)	3 (30)	4 (40)
Be able to identify CDC-recognized programs for diabetes or cardiovascular disease in your area	0 (0)	1 (10)	2 (20)	2 (20)	5 (50)
Advocate for CHW skills and roles in pharmacy settings	0 (0)	1 (10)	1 (10)	2 (20)	6 (60)
As a result of program participation, how likely are you to:					
	Extremely unlikely	Somewhat unlikely	Neither likely nor unlikely	Somewhat likely	Extremely likely
Actively integrate your CHW skills into routine practice in the pharmacy setting	0 (0)	1 (10)	1 (10)	2 (20)	6 (60)
Make patient referrals to CDC-recognized programs for diabetes or cardiovascular disease	0 (0)	1 (10)	1 (10)	1 (10)	7 (70)
Build referral relationships with a CDC-recognized program for diabetes or cardiovascular disease	0 (0)	1 (10)	2 (20)	2 (20)	5 (50)

CDC = Centers for Disease Control and Prevention; CHW = community health worker.

**Table 2 pharmacy-11-00135-t002:** Characteristics of referrals (*n* = 47).

Characteristic	*n* (%)
Program Type	
Pre-diabetes	4 (8.5)
Diabetes	25 (53.2)
Weight management/lifestyle change	9 (19.1)
Other	9 (19.1)
Way referral was made	
Program information provided to patient/patient representative	41 (87.2)
Patient information provided to program	4 (8.5)
Program information provided to patient/patient representative AND patient information provided to program	2 (4.3)

## Data Availability

The data presented in this work are available in Table 1 and Table 2.

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
