# Peer review of "Cross-Training Pharmacy Professionals as Community Health Workers"

_pharmacy, 2023, doi:10.3390/pharmacy11050135_

Round 1

Reviewer 1 Report (New Reviewer)

The manuscript clearly conveys the main topic of the study, which is cross-training pharmacy professionals as community health workers (CHWs). The manuscript follows a structured format.

Suggestions:

Introduction: This section may need to be elaborated to clarify the magnitude of CHWs shortage and how it impacts functionality (referral numbers) in Idaho and/or other states. If the authors could cite some statistics to demonstrate the gap issues, it would strengthen the introduction. It could be best if the authors could provide a conceptual framework of the gaps and solutions. Is it the first initiative to arm pharmacists with CHW skills?

Materials and methods: The rationale behind choosing Idaho to pilot the program should be explained. Additionally, it would be beneficial to provide more details about the program's capacity, including whether 14 participants represent the maximal capacity. More information about the program's structure, curriculum, teaching methods, and specific competencies that need to be achieved should also be included.

Results: In addition to presenting the program evaluations, the abstract could provide more specific findings or feedback received from participants. This could include participant satisfaction levels, areas of improvement, or any challenges faced during the training. Examining whether demographic factors of participants (such as age, sex, years of experience, etc.) are associated with evaluation results or referral outcomes could add valuable insights. It could be best if the authors could include some correlation analysis to explore potential relationships between these variables.

Discussion: If possible, to highlight some challenges and future perspectives of the program, considering similar programs in the world or in the US. What is the importance or paradigm shift that this particular initiative brings to.

Author Response

Reviewer 2 Report (Previous Reviewer 1)

Thank you for the opportunity to review this manuscript which is considerably improved.  As a small scale, pilot study, I believe it now sets clearly the context within the USA and beyond to make the study much more understandable to an international audience.

The major concern is that there there is no mention of ethical review. Whilst I accept that this study may be exempt under local regulations, this needs to be stated.  

The results/discussion would benefit from some comments about the time commitment involved, and whether the pharmacy staff would be likely to have the available time to undertake these additional services going forward.

Round 2

Reviewer 1 Report (New Reviewer)

The authors addressed point by point to the reviewer report and revised the manuscript accordingly.  I have no further comments.

This manuscript is a resubmission of an earlier submission. The following is a list of the peer review reports and author responses from that submission.

Round 1

Reviewer 1 Report

Thank you for the opportunity to review this interesting study.  This is clearly an emerging role, with large variations in qualification and training requirements between states. As a small scale pilot study it adds interesting information but the authors have drawn and appropriate balance between learning from the findings of the study and re-applying them within the local state and legislative framework and recommendations for the wider USA.   For international audiences and wider comparison, it is important to understand the training and background of the individuals involved in the program, as pharmacy technician training and registration requirements vary widely around the globe, but this is adequately explained for international readers to contextualise the findings.  

Reviewer 2 Report

Thank you for the opportunity to review your manuscript. I have included suggestions below.

Abstract:

1.      Consider deleting the following:

a.      “Program participants had their course fee covered and were provided with funds to support their time to complete the course.” It does not add to our understanding of the manuscript and is better suited for the overall manuscript.

b.      “with a financial incentive serving as an effective recruitment tool” At this point it is unclear how a financial incentive is effective given only 14 pharmacy professional participated. Consider deleting this as well.

Manuscript:

1.      While it is not critical, its strange to see the goals of the program “increased referrals to specific chronic disease management programs” and see no emphasis of these programs within the Introduction. Furthermore, since program referrals are also referenced in the Results it is unusual to see no emphasis on these programs in the Introduction.

2.      Please indicate at the appropriate spot IRB related information. Such as the approval number, date the project was approved, and any other relevant information.

3.      Please removal mention of financial incentives in the conclusion. The success of financial incentives is not evaluated based on the data collected.

Reviewer 3 Report

I like the topic and the idea of training pharmacists to be CHW's and I do think this article has potential. That being said, I would like to propose that if the editors choose to publish this, that the formatting and type of article it is be significantly changed (see comments in next paragraph about changing from presenting study findings to having it be more of a commentary). My favorite section in this paper was the discussion and description of how CHW certification is not uniform with widely varying requirements. I would suggest starting the paper with this information, but also include why having pharmacists be certified as CHW's is a good idea.

The way that this article is written is as if it is a study. The goal to “increase number of trained CHW’s and expand referrals” is not a study aim but just the description of a goal for implementing a new program. While it is fine to publish an article that describes how a program was implemented, I would recommend adjusting this article to one that describes what was done. Part of why I don’t think the “results” are considered part of a study is because the baseline requirements for participation were to “provide at least two referrals to CDC-recognized programs.” For 14 participants making 2 referrals each, this should mean that 28 referrals would be made which is what this article reports. However, given that this was a requirement, it was expected that this should happen. I do like the idea of mentioning what kind of referrals were the most common, but this could be discussed as program findings. Even better if made into a table.

SOOO many abbreviations and not all of them are defined before using them. Examples include CDC, and C3. Suggest adding a table in the beginning that defines all the abbreviations.

Having a hard time with what seems like a promotion of the CEWT CHW course as this could indicate a conflict of interest given that the course costs money to take. The description of the course could be used since there are mentions of other certifications in other states; i.e. the course could be compared to these. Along a similar topic, please change “Google Site” to "a website". Saying that it is a Google Site doesn’t add anything.

Line 65: mentions “paid through sustainable mechanisms…” I would like to see a lot more information on what is meant by this. Later there is information provided with how much participants were paid for their time in order to take the course, so wondering if this is what is meant? If so, I would argue that this was a 1 time payment to garner support for participation and wouldn’t be considered “sustainable”. What I think about when I hear that word is reimbursement and payment models for keeping this type of service in the pharmacy (i.e. having a sustainable service).

Line 64: goal is to increase number of trained CHW’s and expand referrals.

Line 79-81: sentence here doesn’t make sense. What are “C3 roles and skills” and what national standards are being mapped to the training?

Table 1: The fact that there were only 5 people who took the course evaluation makes it difficult to draw any meaningful relationships between participants preparedness and the course. There is mention earlier that participants were asked to provide how they wanted to incorporate CHW services into their pharmacy and I think knowing this information would be much more valuable. A table for demographics would also be more helpful than the evaluation responses.

References: I did not do a thorough evaluation of these; however, the number here seems low. I have used the reference about how many people live near a pharmacy, and I can say that what is used to reference this number is not the original reference but one that cites the original study. 
This, along with the low number of references, makes me think that a lot of this happened and the references should be looked at closely.